# Orbital topological edge states and phase transitions in one-dimensional acoustic resonator chains

Feng Gao[1,5], Xiao Xiang[1,5], Yu-Gui Peng ®[1] ✉, Xiang Ni[2,3], Qi-Li Sun[1], Simon Yves[3], Xue-Feng Zhu ®[1] ✉ & Andrea Alù ®[3,4] ✉

Topological phases of matter have attracted significant attention in recent years, due to the unusual robustness of their response to defects and disorder. Various research efforts have been exploring classical and quantum topological wave phenomena in engineered materials, in which different degrees of freedom (DoFs) – for the most part based on broken crystal symmetries associated with pseudo-spins – induce synthetic gauge fields that support topological phases and unveil distinct forms of wave propagation. However, spin is not the only viable option to induce topological effects. Intrinsic orbital DoFs in spinless systems may offer a powerful alternative platform, mostly unexplored to date. Here we reveal orbital-selective wave-matter interactions in acoustic systems supporting multiple orbital DoFs, and report the experimental demonstration of disorder-immune orbital-induced topological edge states in a zigzag acoustic 1D spinless lattice. This work expands the study of topological phases based on orbitals, paving the way to explore other orbital-dependent phenomena in spinless systems.

Topological insulators (TIs) represent phases of matter, and have attracted extensive attention in the past years[1]. Originated in the field of condensed matter, the concept of TIs has been recently transposed to artificial crystals and metamaterials for classical wave systems, demonstrating a plethora of exotic phenomena associated with topological protection, such as reflection-less edge states that are unusually robust to defects and disorder[2–9] and unidirectional, boundary-independent wave transport[10–12]. Topological edge modes, in particular, governed by the bulk-edge correspondence, are localized at the boundaries of nontrivial artificial crystals or at the interface between crystals with different topological phases. Researchers have exploited their robust wave guiding properties to implement exciting functionalities, such as topological lasers[13,14], topological beam splitters[15,16], momentum-locked directional antennas[17,18] and 5 G wireless communication devices[19,20].

Topological insulators are enabled by broken crystal symmetries and associated degrees of freedom (DoFs), such as spin or valley DoFs in a periodic lattice, playing an essential role in inducing various topological phases[1,21]. For example, the quantum-spin Hall effect relies on the interaction of spin and orbital DoFs to induce the Z2 topological insulator[22,23]. Interestingly, a plethora of mechanisms based on DoFs, such as pseudo-spin[2–6] and pseudo-valley spin interactions[7–9], have been very difficult to observe in condensed matter systems, yet they have been successfully implemented in their classical wave analogues in the past decade. In parallel, internal DoFs of a single resonator or a cavity, such as frequency[24–27] and orbital angular momentum[28,29], have been utilized as synthetic dimensions to enrich the topological response for classical waves.

Apart from the degree of spin, the intrinsic orbital degree also plays a crucial part in correlated electrons[30] and solid state materials[31],

[1]School of Physics and Innovation Institute, Huazhong University of Science and Technology, Wuhan 430074, China. [2]School of Physics and Electronics, Central South University, Changsha 410083, China. [3]Photonics Initiative, Advanced Science Research Center, City University of New York, New York, NY 10031, USA. [4]Physics Program, Graduate Center, City University of New York, New York, NY 10016, USA. [5]These authors contributed equally: Feng Gao, Xiao Xiang. ✉e-mail: ygpeng@hust.edu.cn; xfzhu@hust.edu.cn; aalu@gc.cuny.edu

generating many unique topological phases such as orbital superfluidity[32] and topological semimetals[33]. Recently, orbital investigation has been extended to the realms of photonics[13,34–36] and electronic arrays[37,38]. For instance, zigzag-arranged dielectric spherical particles were used to demonstrate electromagnetic topological edge states in microwave experiments[36], and topological lasing in a zigzag array was realized based on edge modes supported by polariton micropillars[13]. In addition, orbital edge states[34], and Type-II/Type-III Dirac cone[35] were experimentally realized using coupled micropillars etched in a photonic honeycomb lattice. By using a scanning tunneling microscope, an analogue of the crystal-field splitting and the $p$ orbital flat band and Dirac cone were also confirmed in electronic Lieb[37] and honeycomb lattices[38], respectively. Finally, intrinsic orbitals have been used to investigate higher-order topological phases[39,40] and valley physics[41]. For classical wave systems, it is worth mentioning that multi-mode plates/resonators and intrinsic degenerate modes interactions also open up plenty of opportunities for engineering topological phase. For instance, paired degenerate plate modes[42] instead of the general single cavity resonances[43–49] were utilized to realize zero-dispersion bands and topological surface phonons. Besides, the interplay between the dipolar and quadrupole modes has been widely harnessed to construct pseudo-spin-dependent topological insulators based on band invesion[4,5].

In this work, we focus on the interaction between degenerate orbitals inside one resonator instead of the interplay between resonator clusters. In the judiciously-designed disk-shaped dimer acoustic cavity unit, different interactions between the degenerate orthogonal

orbitals can be expected, leading to four distinct resonance peaks in the transmission spectra. Based on this platform, we demonstrate orbital-controlled one-dimensional (1D) arrays of coupled cavities as an exemplary model to induce topological acoustic phases. We theoretically illustrate the orbital Su-Schrieffer-Heeger (SSH) model and experimentally demonstrate the orbital-dependent non-trivial topological edge states as well as their robustness against disorders, which is associated with chiral symmetry. We also reveal a duality symmetry in the orbital-induced TIs at different bonding angles and the counterintuitive topological edge states beyond the conventional SSH lattices. Our work lays a foundation for investigating classical wave interactions among multiple orbitals and the associated topological phase transition, and provides a prospective opportunity for further explorations involving orbital-dependent devices in various wave platforms.

## Results

### Orbital interactions

In quantum mechanics, the hydrogen atom has discrete energy bands of high symmetries, for example the one-fold $s$ band and three-fold $p$ bands ($p_x$, $p_y$, $p_z$). As sketched in Fig. 1a, the wave functions are spherically symmetric for the $s$ orbital and have different directionality in space for $p$ orbitals. In an acoustic cavity (or a meta-atom), the local resonances can mimic the discrete energy bands in atoms, where the $s$ orbital eigenmode cannot be easily excited due to its nearly zero eigen-frequency. Therefore, the first-order Fabry-Pèrot mode is typically chosen as the resonant mode of interest in

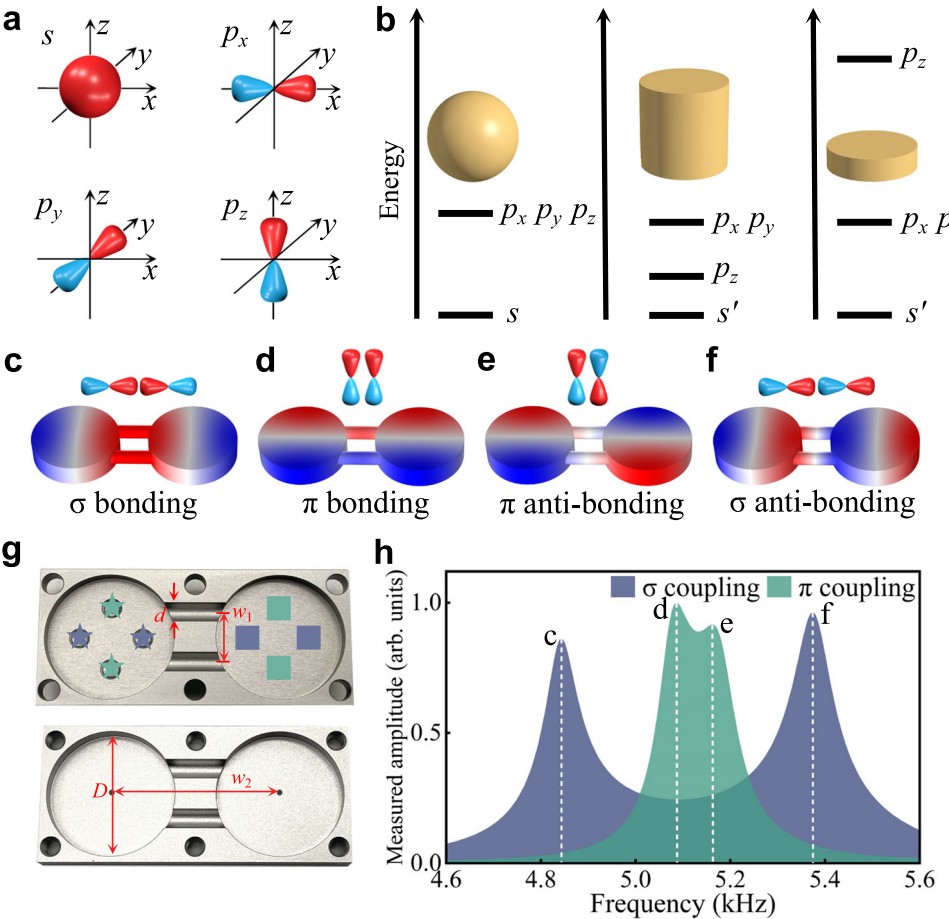

**Fig. 1 | Orbital interactions in an acoustic dimer meta-atom. a** Illustration of different orbitals. **b** Three acoustic meta-atoms and the related discrete energy levels. **c**–**f** Schematics and spatial field distributions of four distinct eigenmodes for the designed dimer unit obtained from numerical calculations. **g** Pictures of the fabricated sample. Stars and squares mark the excitation and measurement sites, respectively. **h** Measured transmission responses for different orbital excitations. Source data are provided as a Source Data file.

previously studied topological insulators[43–49]. However, more complex meta-atom geometries may support a wider range of discrete energy spectra, as displayed for three canonical geometries in Fig. 1b. Similar to the hydrogen atom, spherical resonators can support three degenerate orbital resonances, associated to the three dipole modes, $p_x$, $p_y$, $p_z$. The degeneracy of $p$ resonances is lifted when the spherical symmetry is broken. For a cylindrical resonator, the $p_z$ orbital has a lower frequency than the $p_x$ and $p_y$ orbitals, which can result in hybrid excitations of $p_x$, $p_y$, $p_z$ orbitals if the relevant resonances are not too far away. In a disk resonator, on the contrary, the $p_z$ orbital has much higher resonance frequency than the $p_x$ and $p_y$ orbitals, and the in-plane $p_x$ and $p_y$ orbitals can be specifically excited and controlled through the associated pair of intrinsic DoFs. This is our geometry of choice to define orbital-induced topological phase in a 1-D array of these meta-atoms.

We start by discussing the coupling between two identical disks. As shown in Fig. 1c–f, there are two types of couplings between the in-plane $p$ orbitals: $\sigma$ bonding and $\pi$ bonding. Specifically, $\sigma$ bonding describes the coupling between the meta-atoms with orbital orientation parallel to the bonding direction, while $\pi$ bonding describes the hopping for orbital orientation normal to the bonding direction.

In Fig. 1c–f, we show the four different eigenmodes associated with the interacting in-plane $p$ orbitals, which are $\sigma$ bonding, $\pi$ bonding, $\pi$ anti-bonding, and $\sigma$ anti-bonding, respectively, displayed with increasing eigenfrequencies. Here, the bonding and anti-bonding modes are featured with $p$ orbitals in the coupling sites of symmetric and anti-symmetric field distributions, respectively, with respect to the central axis of the dimer unit. The frequency difference between the anti-bonding and bonding modes is proportional to the coupling strength of $\sigma$ or $\pi$ bonding. It is intuitive that the coupling strength of $\pi$ bonding is smaller than the one of $\sigma$ bonding, owing to less wave-function overlap. The coupling strength can be precisely tailored by the geometric parameters of the coupling channels.

A fabricated sample of dimer meta-atom is shown in Fig. 1g. The dimer unit comprises two pieces of processed aluminum plates, while the diameters of the coupling tubes and cavities are $d = 6\,\text{mm}$ and $D = 40\,\text{mm}$, respectively. The distances between the coupling tubes and cavities are $w_1 = 16\,\text{mm}$ and $w_2 = 53\,\text{mm}$. In order to experimentally excite and measure the pressure amplitude spectra and field profiles, each site resonator was processed with four holes for sound wave excitation and detection. The holes were sealed with screws when not in operation, and were used for the insertion of speakers and microphones in the measurements. The amplitude responses for the $\sigma$ ($\pi$) couplings were measured by putting speakers into the holes marked by the dark blue (green) stars, and inserting microphones into the holes marked by the dark blue (green) squares, as shown in Fig. 1g. The experimental results displayed in Fig. 1h, featuring four resonance peaks labelled as c, d, e, f in the normalized spectra, match well with the theoretical model in Fig. 1c–f. The asymmetry in the resonance peaks is caused by small asymmetric sound leakages in the measurements. Moreover, we display the measured phase of the retrieved pressure field, further verifying the excitation of the different $p$ orbitals, in good agreement with our theoretical and numerical predictions (see Fig. S1 in Supporting Information). The experiment confirms the realization of coupled orbitals in acoustics and verifies the emergence of complex interactions between $p$-orbital modes.

## Orbital-induced topological transitions

By periodically coupling the dimer unit cell into a 1D array, we can obtain a bonding-angle dependent 1D acoustic array, in which the bonding-angle is defined between two identical coupling waveguides, as shown in Fig. 2a. Within the tight-binding approximation[34,35], the orbital-dependent lattice can be characterized by the real-space

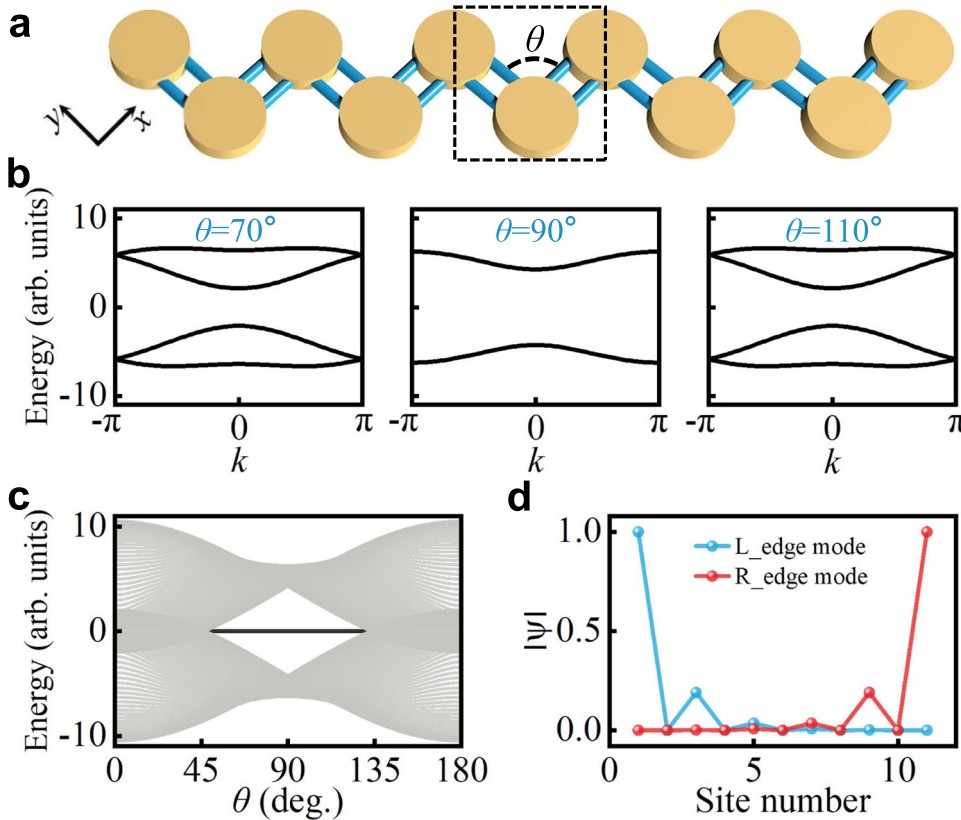

**Fig. 2 | The orbital SSH model with topological edge states. a** Illustration of the orbital SSH model. **b** Calculated band structures for different bonding angles. **c** The energy spectra for a finite array. The bold line in the central region denotes the emergence of topological edge states. **d** Edge mode profiles for $\theta_c = 90°$.

Hamiltonian

$$H = \sum_{n,i,j}(t_{ij}a^\dagger_{n,i}b_{n,j} + t'_{ij}b^\dagger_{n-1,i}a_{n,j}) + h.c.,\tag{1}$$

where $a^\dagger_{n,i}$ ($a_{n,i}$) is the creation (annihilation) operator for the resonant modes associated with $p$ orbitals of indices $i(j) = \sigma,\pi$, representing the orientation directions in the sublattice site $A_n$, with the index $n$ labeling the unit-cell number in the array. $t_{ij}(t'_{ij})$ represents the hopping strength between the respective modes (see derivation details in Supplementary Note 3). In order to calculate the band structure and topological invariant in the orbital-dependent 1D model, we take the Fourier transformation for the real basis of the Hamiltonian in Eq. (1). By harnessing the spinor vector $\psi = [a_{\mathbf{k},x}, a_{\mathbf{k},y}, b_{\mathbf{k},x}, b_{\mathbf{k},y}]^T$, the systematic Hamiltonian with discretized momenta can be described by $H = \sum_{\mathbf{k}}\psi^\dagger H(\mathbf{k})\psi$. Specifically, the matrix $H(\mathbf{k})$ reads

$$H(\mathbf{k}) = \begin{pmatrix} 0 & D \\ D^\dagger & 0 \end{pmatrix}, D = \begin{pmatrix} D_1 & D_2 \\ D_3 & D_4 \end{pmatrix},\tag{2}$$

in which the elements of the $2 \times 2$ matrix $D$ are

$$D_1 = t_\pi + [t_\sigma\sin^2(\theta) + t_\pi\cos^2(\theta)]e^{i\mathbf{k}},$$

$$D_2 = D_3 = (t_\pi - t_\sigma)\sin(\theta)\cos(\theta)e^{i\mathbf{k}},$$

$$D_4 = t_\sigma + [t_\pi\sin^2(\theta) + t_\sigma\cos^2(\theta)]e^{i\mathbf{k}},$$

where $t_\sigma$ and $t_\pi$ denote the hopping strengths longitudinal and transverse to the bond between the cavities, respectively, and $\mathbf{k}$ is the Bloch momentum along the periodic direction. In a zigzag chain with a bonding angle /theta displayed in Fig. 2a, the acoustic wave couplings are subjected to alternating strengths along the chain for $p_x$ or $p_y$ orbital. Each subspace corresponds to one copy of the conventional SSH model, and it obeys chiral symmetry. Therefore, we refer to our model as an orbital-dependent SSH model.

On the other hand, if symmetries such as space-time symmetry or internal symmetries of the system are preserved, their corresponding operators commute with the Hamiltonian, making the physical system invariant under the symmetry transformation. Here, we present a duality operator[50,51], operating in the same way as a symmetry operator, which changes the parameters in a certain pattern to map one system into another one with the same band structure. Concretely, in Fig. 2b, the left band structure for $\theta = 70°$ and the right band structure for $\theta = 110°$ are dual to each other, while the middle band structure for $\theta_c = 90°$ is two-fold degenerate and self-dual[50,51]. In fact, any orbital lattice with hybrid bonding angles $\theta$ and $2\theta_c - \theta$ possesses the same energy spectra, due to the hidden duality symmetry. Therefore, our two-DoFs orbital SSH model provides an excellent platform to investigate the mathematical mapping between different lattices and the duality beyond the conventional SSH model. (see additional details on the duality Hamiltonian in Supplementary Note 4).

The energy spectra for the finite orbital SSH model at various bonding angles are shown in Fig. 2c. The highlighted band in the center gap denotes nontrivial edge states. The mode profiles at $\theta_c = 90°$ are shown in Fig. 2d, where the two orbital modes are strongly localized at the leftmost and rightmost sites, respectively. For the bonding angles close to $\theta = 0°$ and $\theta = 180°$, this orbital lattice is topologically trivial and has no bandgaps. In our sample, the bonding angle cannot be smaller than 45° to be mapped into our model, and the described phenomena can be also explored in alternative samples with bonding angles between 135° and 180° due to the hidden duality symmetry. Particularly, for $\theta = 180°$, the zigzag chain turns into a straight one, in which the two intrinsic orbitals become decoupled and the bandgaps

are closed (see Fig. S9 in Supplementary Information). Moreover, we show that a topological phase transition arises at $|\theta - \pi/2| = \arcsin|(\gamma+1)/(\gamma-1)|$, $\gamma = t_\sigma/t_\pi$, indicating that the transition point depends on the ratio between longitudinal and transverse hopping strengths (see details in Supplementary Note 3). The winding number charactering the topological properties of the Hamiltonian can be defined as[36]

$$\mathcal{W} = \frac{i}{2\pi}\int_{-\pi}^{\pi}dk\frac{d\ln\det D(k)}{dk} = -\frac{1}{2\pi}\int_{\mathcal{C}}d\arg\det D(k)\tag{3}$$

where $\mathcal{C}$ represents a contour swept by $D(k)$ as $k$ varies across the Brillouin zone (see Supplementary Note 3).

## Experimental demonstration

The emergence of topologically protected orbital-induced edge states in the acoustic lattice was validated in experiments. In Fig. 3a, we exhibit the realized sample, which comprises 11 identical disk resonators connected by coupling channels with a bonding angle of $\theta = 90°$. In this specific configuration, the two in-plane $p$ orbitals in each cavity are orthogonal to each other. The calculated eigenfrequencies in the orbital SSH lattice are shown in the left panel of Fig. 3b. Two degenerate nontrivial edge states appear at around 5026 Hz in the bandgap, as labeled by the blue and red spheres, where the insets show the corresponding simulations of left and right edge states.

We conducted four separate measurements with two different orbital sources in the site resonators close to the left and right edges, as shown in Fig. 3a, in which the orbital sources, referred as L_$p_x$ (R_$p_x$) and L_$p_y$ (R_$p_y$) and marked by solid and dashed arrows, consist of two acoustic sources which are out of phase and set to be parallel and perpendicular to coupling waveguides, respectively. The measured pressure amplitude response on the edge-site resonators are displayed in the middle and right panels of Fig. 3b. As expected, the measured pressure amplitude spectrum at the left (right) end site for the L_$p_x$ (R_$p_y$) source excitation has a prominent peak at around 5101 Hz in the bandgap region. The frequency of the spectrum peak has a slight deviation from the simulated one due to inevitable loss in the experiment. On the other hand, for the L_$p_y$ (R_$p_x$) excitation, the peaks in the bandgap vanish.

To clearly unveil this orbital-dependent topological feature, we further scanned and mapped out the intensity fields in the lattice at the peak frequency for four different excitations. As shown in Fig. 3c, d, sound waves were strongly localized at the left (right) end-site resonator for L_$p_x$ (R_$p_y$) excitations, and spreaded into the bulk for L_$p_y$ (R_$p_x$) excitations. Only excitations coupling to in-plane dipolar orbitals can efficiently excite the orbital-dependent topological edge states, whereas a monopole source does not work due to symmetry mismatch (see Supplementary Note 6). Our experimental results univocally reveal the observation of orbital-induced topological edge states.

## Disorder robustness

We further investigated the robustness of the orbital-induced topological edge states against structural disorder. As schematically shown in Fig. 4a, in a set of experiments the bonding angles are made no longer identical but have a random distribution within the range [70°, 110°]. The random bonding angles between consecutive coupling links are numbered as $\theta_i$ ($i = 1, 2, ...9$) in order. As displayed in Fig. 4b, we fabricated one aperiodic sample with varying bonding angles to check the robustness. The energy stability of topological edge modes against disorder in bonding angles is confirmed by the calculated energy spectra for 50 random cases in Fig. 4c. The set of bonding angles in the fabricated sample is the case marked by the cyan arrow in Fig. 4c. Clearly, for the 50 considered random geometries, the topological

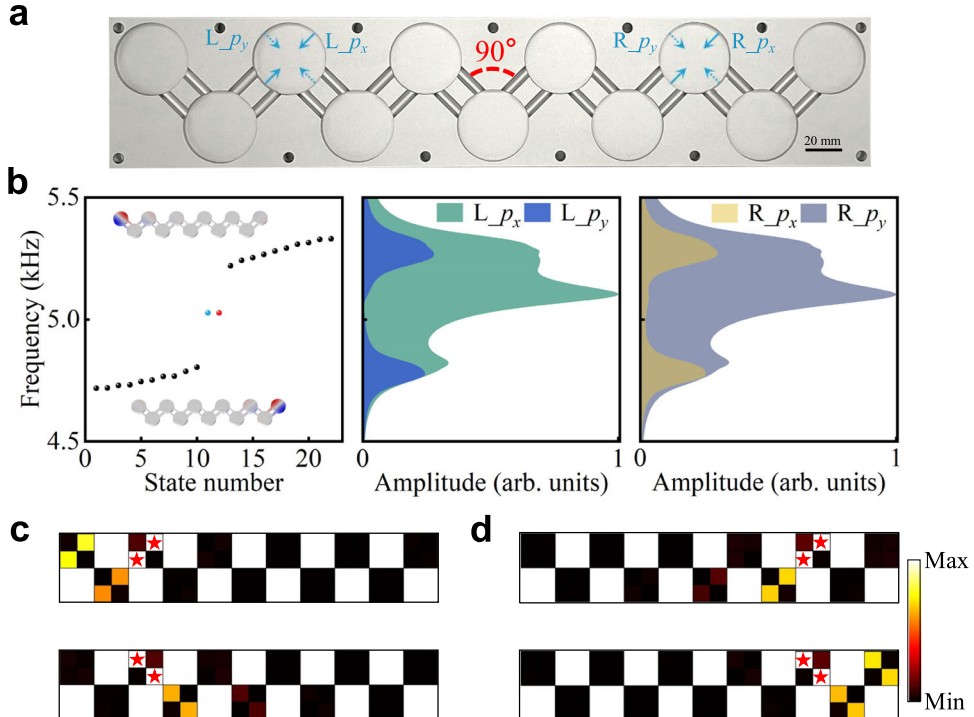

**Fig. 3 | Orbital-induced topological edge states. a** Fabricated sample. **b** Left: Calculated eigenvalue spectrum for a finite acoustic lattice with 11 cavities and simulations of topological edge states. Middle: Measured amplitude spectra in the left edge-site resonator for the L_$p_x$ and L_$p_y$ excitations respectively, as marked by the solid and dashed arrows in (**a**). Right: Measured amplitude spectra at the right edge-site resonator for the R_$p_x$ and R_$p_y$ excitations as marked by the solid and dashed arrows in (**a**). **c**, **d** Measured pressure amplitude field distributions for the four different excitations in (**a**) and (**b**). Source data are provided as a Source Data file.

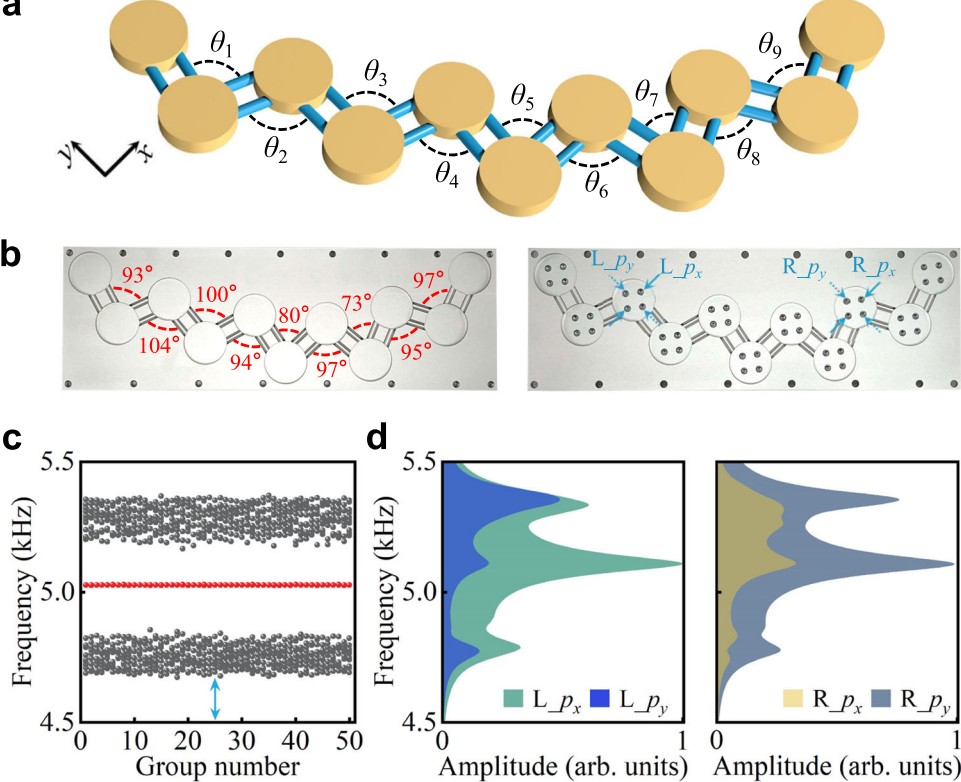

**Fig. 4 | Disorder robustness of orbital-induced TI. a** Illustration of an aperiodic orbital lattice. **b** Fabricated sample. **c** Energy spectra for 50 different aperiodic lattices with the randomly chosen bonding angles. **d** Measured amplitude spectra for different orbital excitations. Source data are provided as a Source Data file.

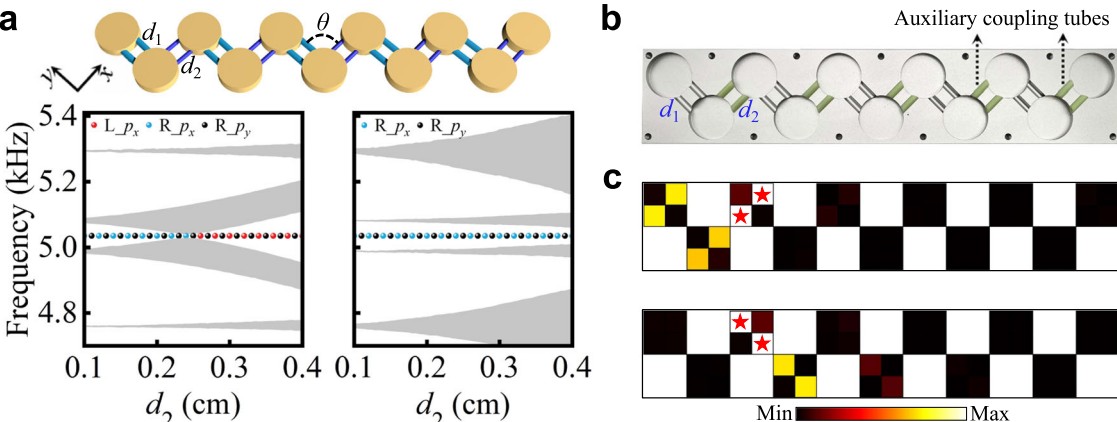

**Fig. 5 | Topological transitions and edge modes in dimerized orbital SSH model. a** Top: Sketch of the orbital SSH model with dimerized coupling tubes. Bottom: Energy spectra for orbital SSH models and conventional SSH models by varying $d_2$ from 0.1 cm to 0.4 cm. **b** Experimental sample with auxiliary coupling tubes. **c** Measured pressure amplitude fields for the L_$p_x$ and L_$p_y$ excitations. Source data are provided as a Source Data file.

edge states always survive and remain pinned at the zero energy (red spheres in the bandgap) because of their topological nature.

To further verify the disorder robustness of orbital-induced TIs, we measured the pressure amplitude spectra for different orbital excitations. The orientations of orbital sources are depicted in the right panel of Fig. 4b. We conducted four frequency-resolved acoustic measurements on the edge-site resonators for L_$p_x$, L_$p_y$, R_$p_x$, and R_$p_y$ orbital sources, with results summarized in Fig. 4d. Prominent peaks still emerge in the bandgap for L_$p_x$ and R_$p_y$ excitations at around 5108 Hz, similar to the results for the periodic chain in Fig. 3b. The orbital-induced topological properties are hence preserved and immune against the bonding-angle disorder. We show the detailed information on the calculated transmission spectra and the measured pressure field distributions in this 1D disordered orbital lattice (see Supplementary Note 7).

## Beyond conventional SSH model

In order to visualize the generality of topological phase transitions emerging in our orbital SSH model, we furtherly dimerize the hopping amplitudes by varying the diameter of the coupling tubes, as shown in Fig. 5a. For the leftmost and rightmost resonators, they are strongly and weakly coupled to the topological chain with different coupling tubes $d_1$ and $d_2$ ($d_2 < d_1 = 0.6$ cm), respectively. Unlike the conventional SSH model, with edge modes residing on the rightmost resonator, topological phase transitions emerge in our orbital SSH chains. Keeping the bonding angle at 90°, the $p_x$-like edge mode displays tunneling through the bulk chain from the right to the left termination by varying $d_2$ from 0.1 cm to 0.4 cm, and the transition appears at $d_2 = 0.25$ cm in the simulations. This topological transition is not found in the conventional counterpart obtained by setting the bonding angle as 180°, as seen in Fig. 5a. The $p_x$-like modes at the left and right ends are denoted as red and light-blue spheres, respectively. With 3D-printed auxiliary tubes, we adapt the sample with identical coupling tubes in Fig. 3a to form the dimerized one shown in Fig. 5b with $d_2 = 0.4$ cm. In experiments, the counterintuitive edge mode on left-most resonator is evidenced with a $p_x$-like source, as shown in Fig. 5c, agreeing well with the simulations. Meanwhile, a $p_y$-like source cannot excite the leftmost resonator, featuring strong orbital-selectivity.

## Discussion

In summary, we have proposed a strategy to enable tailored wave interactions between multiple orbitals in acoustic discrete systems, and experimentally demonstrated nontrivial orbital-induced topological insulators in a spinless SSH model. We also unveiled a hidden duality symmetry, and topological phase transitions in this system. In experiments, we demonstrated the close correlation between the excitation orientation and orbital-dependent edge modes, which agrees well with numerical predictions. In addition, we demonstrated strong topological protection against bonding-angle disorders, and counterintuitive topological edge modes in orbital SSH lattices beyond the response in conventional SSH arrays. Our work extends one-DoF topological insulators to the two-DoFs orbital regime, providing a route for exploring orbital-dependent topological physics and functional devices.

## Methods
### Numerical simulations
Full-wave numerical simulations were implemented by harnessing a finite element solver. The walls of acoustic resonators and connecting waveguides are assumed rigid in the simulations, owing to the large acoustic impedance mismatch between (metal Aluminum, photo-sensitive resin) and air. The mass density and speed of sound in air are assumed to be $\rho_{air} = 1.29$ kg/m$^3$ and $c_{air} = 343$ m/s. When calculating the bulk band in Fig. 2b, and S3, the Bloch boundary condition is employed for the periodic orbital orientation.

### Experiments
The designed samples were manufactured by means of metal machining technique and 3D printing (geometry tolerance of 0.1 mm). To facilitate the sound excitation and detection, four holes with the radius of ~2.5 mm were drilled on the top of each disk cavity. In experiments, a pair of out-of-phase sound signals were launched inside the excitation acoustic cavity, as shown in Fig. 3 and Fig. 4. A 1/8 inch microphone (Brüel & Kjær 4138-A-15) was employed for detecting the amplitude and phase of sound waves in each disk cavity, accompanied with the other one in the same cavity as phase reference. The sound signals, recorded and processed via a network analyzer (Brüel & Kjær 3160-A-042), were utilized to map out the pressure amplitude spectra as well as the amplitude fields in the lattices.

## Data availability
The main data supporting the findings of this study are available within this letter and its supplementary information. The source data generated in this study have been deposited in Figshare repository https://doi.org/10.6084/m9.figshare.24581112. Source data are provided in this paper.

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

## Acknowledgements

We acknowledge the constructive discussions with Prof. Rui Yu in the department of physics of Wuhan University. This work was supported by fundings from the National Natural Science Foundation of China (Nos. 12304492 (Y.-G.P.), 11690030 (X.-F.Z.), and 11690032 (X.-F.Z.)). X. Ni, S. Yves and A. Alù were supported by the Simons Foundation and AFOSR Foundation.

## Author contributions

F.G. and X.X. contributed equally to this work. Y.-G.P., X.-F.Z., and A.A. conceived the idea and supervised the project. F.G., Y.-G.P., X.N., and Q.-L.S. developed the theory and did the simulations. F.G. and Y.-G.P. designed the experiments and fabricated the samples. F.G. and X.X. performed the experiments. F.G., Y.-G.P., S.Y., X.-F.Z., and A.A. contributed to the writing of the manuscript. All authors analyzed the data.

## Competing interests

The authors declare no competing interests.
