## [Peer Review File · Nature Communications]

Orbital topological edge states and phase transitions in one-dimensional acoustic resonator chainsREVIEWER COMMENTS

Reviewer #2 (Remarks to the Author):

This work reports the experimental realization of a 1D acoustic topological system with protected edge modes. The key selling point is the use of the orbital degree of freedom (p_x and p_y orbitals), which makes the system different from conventional acoustic SSH systems where only a single orbital is utilized. The authors demonstrate how orbital interaction can induce a coupling dimerization even when all coupling channels are identical, and they also show in fig. 5 with unequal d_1 and d_2 the system can behave differently from the conventional SSH model. I think these results prove that orbital degree of freedom can indeed bring us richer topological physics. In general, the manuscript is well-written, and the experimental data look solid. However, I still have several questions for the authors to address before I can recommend its publication.

(1) In the introduction part, the authors treat spin and orbital as two different degrees of freedom. This is true for electronic systems. However, for acoustics (and other classical waves), the orbital degree of freedom is sometimes used to construct pseudo spin (for example, in the classical quantum spin Hall phase). I think this should be mentioned to avoid potential confusion.

(2) Similar models have been studied in photonics (e.g., [Nature Photonics, 11, 651 (2017)] and [PRL 114, 123901 (2015)]). These previous developments should be acknowledged.

(3) There are several other interesting uses of the orbital degree of freedom in creating topological phases: [Nature Communications 13, 6597 (2022)] and [arXiv:2110.05703 (2021)].

(4) Calculation of the topological invariant is missing.

(5) Fig. 2c is kind of misleading. I understand that it is obtained from the tight-binding model. But for the realistic structure the angle θ cannot be too small. The authors may indicate the smallest value of θ for the acoustic structure in Fig. 2c.

(6) The dual symmetry looks interesting. In addition to its influence on energy dispersion, can it have other physical consequences?

Reviewer #3 (Remarks to the Author):

In their manuscript "Orbital-Induced Acoustic Topological Order", Gao and collaborators investigate a one dimensional acoustic system of two coupled SSH chains. They realise this system with coupled acoustic resonators which support two modes, which are, in the absence of the coupling, degenerate. This allows to observe physics which is beyond the standard SSH model, i.e., results which are certainly worthy of publication.

I have one main criticism on the presentation of the work which may influence the choice of the optimal journal for this manuscript. I find the physics observed interesting, albeit not exposed in enough detail, I comment on this below. What bothers me more is the claim to introduce a new paradigm for the design of interesting meta material band structures. The use of orbital degrees of freedom has always been a guiding principle to obtain the sought after features. Also in publications of some of the present authors. Having a mode with a node is a standard way to influence the (weak) coupling between coupled such modes. Moreover, using degenerate modes to engineer more complex band structures with non-trivial orbital content is also not novel. For example in Nature Mat. 17, 323 (2018) a detailed recipe is given how two engineer tight-binding models exploiting orbital degrees of freedom, including the use degenerate pairs. In that sense, the current paper is an excellent illustration of these orbital based engineering, rather than a novel avenue. The fact that the above Nature Materials is formulated for waves in thin plates is irrelevant for the novelty. Both thin plates and acoustic cavities are described by classical wave equations.

I would like to stress, however, that I consider the implemented model and observed phenomena of very high interest and novel. Whether this alone warrants a publication in Nature communications, I let the editors decide. My hunch would be that it is not quite ground-breaking enough for a journal like Nature Physics or Nature Materials, but may very well fit here.

Before I can recommend the paper for publication I would invite the authors to comment on the following points:

- 1.) Language: I find the term "topological order" unfortunate. In electron physics it is

reserved for system with long-range entanglement like the fractional quantum Hall effect or certain spin liquids and not band effects. I would therefore avoid the use of "order" also in classical materials.

2.) Language: In line 39 (and elsewhere), the authors use the term "symmetry-breaking" DOF. I don't understand this term. Which symmetry is broken?

3.) Line 59 and 62-64 are not quite correct. Not only have higher-order resonances been considered before (e.g. mode 21 and 21 in the above mentioned Nature Mat.), but a much more detailed analysis of the nearby modes, their influence on the reduced order models and an analysis of longer-range couplings, etc. I would simply remove these claims and devote more to the introduction of the actually studied model.

4.) In Fig. 1h one can see that the mean frequency of the sigma-coupled modes is a hunch lower than those of the pi-coupled bands, indicating that the coupling tubes between the cavities are not only acting as couplers but also influence them modes themselves. A careful analysis of this could be presented (not critical, however).

5.) Maybe a comment on the influence on the coupling strength of a shift of the two couples towards the node could be interesting.

6.) Topology: The authors keep claiming these edge states to be topological. A careful analysis of this is certainly required to make such claims. Which is protecting symmetry? If SSH, it will be some chiral one. How is it represented in the space of the two modes in the two inequivalent cavities? How does this lead to a quantised Berry phase, or alike? Can I go to some block-off-diagonal representation as usual in this symmetry class? What is the influence of the duality on all of this?

Response Letter to Reviewers

We are grateful to all reviewers for their insightful and constructive comments on our manuscript (NCOMMS-23-10871). In the following, the comments from the reviewers are quoted in *blue italic font*, followed by our point-to-point responses. To address these comments, we have substantially revised the manuscript and the supplementary information, and the corresponding updates are highlighted in **red** in the respective files.

Reviewer #2

Reviewer comments:

This work reports the experimental realization of a 1D acoustic topological system with protected edge modes. The key selling point is the use of the orbital degree of freedom (p_x and p_y orbitals), which makes the system different from conventional acoustic SSH systems where only a single orbital is utilized. The authors demonstrate how orbital interaction can induce a coupling dimerization even when all coupling channels are identical, and they also show in fig. 5 with unequal d_1 and d_2 the system can behave differently from the conventional SSH model. I think these results prove that orbital degree of freedom can indeed bring us richer topological physics. In general, the manuscript is well-written, and the experimental data look solid. However, I still have several questions for the authors to address before I can recommend its publication.

Response:

We thank Reviewer #2 for these positive comments and recommendation. Below we provide a point-to-point response to their concerns and comments. We have revised the manuscript and supplementary information following their suggestions.

Reviewer comments:

(1) In the introduction part, the authors treat spin and orbital as two different degrees of freedom. This is true for electronic systems. However, for acoustics (and other classical waves), the orbital degree of freedom is sometimes used to construct pseudo

spin (for example, in the classical quantum spin Hall phase). I think this should be mentioned to avoid potential confusion.

Response:

We thank Reviewer #2 for this constructive suggestion. In this work, we focus on the topological physics enabled by cavity orbitals, namely degenerate orbitals inside one cavity. Researchers also utilize cavity or scatterer clusters to generate different types of orbitals, such as p and d orbitals, as extensively studied in various classical wave topological systems.

In order to avoid possible confusion, we have revised the related sentences in **Lines 76-79 (page 4)** in the *revised manuscript* as “Here, we focus on the interaction between degenerate orbitals inside one resonator instead of the interplay between resonator clusters, which has been utilized to construct pseudo-spin dependent topological insulators for light and sound based on band inversion^{4,5}.”

Updated references:

4. Wu, L.-H. & Hu, X. Scheme for Achieving a Topological Photonic Crystal by Using Dielectric Material. *Phys. Rev. Lett.* **114**, 223901 (2015).

5. He, C. et al. Acoustic topological insulator and robust one-way sound transport. *Nat. Phys.* **12**, 1124-1129 (2016).

Reviewer comments:

(2) Similar models have been studied in photonics (e.g., [Nature Photonics, 11, 651 (2017)] and [PRL 114, 123901 (2015)]). These previous developments should be acknowledged.

Response:

We thank Reviewer #2 for highlighting these high-quality works. We agree that these two references study related configurations in photonics, and demonstrate orbital edge modes. We have cited these works in the revised introduction in **Lines 60-62 (pages 3, 4)** as “with zigzag-arranged dielectric spherical particles, electromagnetic topological

edge states were observed at microwaves³⁶, and topological lasing in a zigzag array with polariton micropillars was realized based on edge modes¹³.”

Updated references:

13. St-Jean, P. et al. Lasing in topological edge states of a one-dimensional lattice. *Nat. Photon.* **11**, 651-656 (2017).

36. Slobozhanyuk, A. P., Poddubny, A. N., Miroshnichenko, A. E., Belov, P. A. & Kivshar, Y. S. Subwavelength topological edge States in optically resonant dielectric structures. *Phys. Rev. Lett.* **114**, 123901 (2015).

Reviewer comments:

(3) *There are several other interesting uses of the orbital degree of freedom in creating topological phases: [Nature Communications 13, 6597 (2022)] and [arXiv:2110.05703 (2021)].*

Response:

We apologize for not highlighting these related works. In [arXiv:2110.05703 (2021)] (now published in *Physical Review Letters*), a parity-dependent higher-order topological state was observed using a Kagome metasurface with designer-plasmonic metaatoms supporting different plasmonic orbitals. In [Nat. Commun. **13**, 6597 (2022)], a photonic quadrupole topological insulator was constructed with orbital-induced synthetic flux via orbital hybridization. These are nice examples demonstrating the power of orbitals in plasmonics and photonics.

We have cited these two demonstrations of higher-order topology in **Lines 67-68 (page 4)** of the revised manuscript: “intrinsic orbitals were also used to investigate higher-order topological phases^{39,40} and valley physics.”

Updated references:

39. Li, Y. et al. Parity Splitting and Polarized-Illumination Selection of Plasmonic Higher-Order Topological States, Preprint at <https://arxiv.org/abs/2110.05703> (2021).

40. Schulz, J., Noh, J., Benalcazar, W. A., Bahl, G. & von Freymann, G. Photonic quadrupole topological insulator using orbital-induced synthetic flux. *Nat. Commun.* **13**, 6597 (2022).

Reviewer comments:

(4) *Calculation of the topological invariant is missing.*

Response:

We thank Reviewer #2 for pointing out this important issue. We have added Eq. (3) in the revised paper to define the topological invariant for our orbital SSH model, as shown in **Lines 216-219 (page 10)** of the revised manuscript: “**The winding number charactering the topological properties of the Hamiltonian can be defined as**³⁶

$$\mathcal{W} = \frac{i}{2\pi} \int_{-\pi}^{\pi} dk \frac{d \ln \det D(k)}{dk} = -\frac{1}{2\pi} \int_C d \arg \det D(k), \quad (3)$$

where C represents a contour swept by $D(k)$ as k varies across the Brillouin zone”

We have also added more details on how to calculate the topological invariant in “**Supplementary Note 3**” of the revised Supplementary Information.

Related content in **Supplementary Note 3** of the revised Supplementary Information: The topological properties of the orbital Hamiltonian are characterized by a winding number defined by⁴

$$\mathcal{W} = \frac{i}{2\pi} \int_{-\pi}^{\pi} dk \frac{d \ln \det D(k)}{dk} = -\frac{1}{2\pi} \int_C d \arg \det D(k) \quad (S16)$$

where C represents a contour swept by $D(k)$ as k varies across the Brillouin zone. For $|\theta - \pi/2| < \arcsin |(\gamma+1)/(\gamma-1)|$ where the system is gapped as shown in Fig. S5a, the contour of $\det D$ forms one turn around zero point, as shown in Fig. S5d, and thus we obtain the topological invariant $\mathcal{W}=1$. For the critical condition at $|\theta - \pi/2| = \arcsin |(\gamma+1)/(\gamma-1)|$, the contour for $\det D$ is displayed in Fig. S5e, and the winding number is ill-defined. When the band is gapless as shown in Fig. S5c, $\det D$ actually passes twice through zero point defined by $\det D = 0$, which is shown in

Fig. S5f. To formally define the winding number, the contour in Fig. S5f can be shifted by an infinitesimal amount to the left or to the right, for which we obtain the topological invariant $\mathcal{W} = 0$.

Fig. S5. Calculated and simulated energy spectra marked by black lines and red spheres for different bonding angles at (a) 100° , (b) 132.9° , and (c) 165° . Schematic sketches for a contour C in the complex plane that $\det D$ follows as k changes from $-\pi$ to π . (d) Scenario for $|\theta - \pi/2| < \arcsin |(\gamma+1)/(\gamma-1)|$. (e) Scenario for $|\theta - \pi/2| = \arcsin |(\gamma+1)/(\gamma-1)|$. (f) Scenario for $|\theta - \pi/2| > \arcsin |(\gamma+1)/(\gamma-1)|$.

Updated references:

36. Slobozhanyuk, A. P., Poddubny, A. N., Miroschnichenko, A. E., Belov, P. A. & Kivshar, Y. S. Subwavelength topological edge States in optically resonant dielectric structures. *Phys. Rev. Lett.* **114**, 123901 (2015).

Reviewer comment:

(5) Fig. 2c is kind of misleading. I understand that it is obtained from the tight-binding model. But for the realistic structure the angle θ cannot be too small. The authors may indicate the smallest value of θ for the acoustic structure in Fig. 2c.

Response:

We thank Reviewer #2 for this insightful suggestion. Yes, the energy spectra shown in Fig. 2c are obtained based on the tight-binding model for a finite array. As mentioned by the reviewer, the bonding angle θ cannot be very small in a practical implementation. Following the reviewer's suggestion, to explicitly illustrate this point we performed simulations to obtain the energy spectra for a realistic zigzag array using COMSOL, and we found that the smallest acceptable angle to be 45° .

To clarify this issue, we have added in the revised manuscript the remark “**In our sample, the bonding angle cannot be smaller than 45° to preserve the connection with our SSH model, and the phenomena described here can be unveiled using bonding angles between 135° and 180° due to the hidden duality symmetry.**” in Lines 209-211 (page 9).

Inspired by this comment, we have also added an alternative design based on a quasi-two-dimensional acoustic array, in which arbitrary in-plane bonding angles can be used, as shown in “**Supplementary Note 5**”.

Related content in **Supplementary Note 5** in the revised Supplementary Information: The investigated zigzag chain model with bonding angle θ is depicted in the top subfigure of Fig. S9a. The smallest accessible θ is 45° in our sample, which is in the trivial regime, and the corresponding eigen-field is shown in the middle subfigure of Fig. S9a. In the bottom subfigure of Fig. S9a, we display the eigen-field for another trivial case of $\theta = 180^\circ$. To avoid any possible confusion, we further plot the energy spectra for the cases with achievable bonding angles in the realistic zigzag array, and the ones with unachievable bonding angles are denoted by a grey rectangle as shown in Fig. S9b. Due to the hidden duality symmetry, the physics of these unachievable cases are identical to the ones with large bonding angles between 135° and 180° .

On the other hand, we can alternatively design another quasi-two-dimensional array as shown Fig. S9c, where arbitrary in-plane bonding angles can be obtained by projection. The unachievable configuration with bonding angle $\theta = 30^\circ$ is depicted as an example in COMSOL. The projective structure in x - y plane is illustrated in Fig. S9d.

The corresponding eigen-fields at the central frequency are also presented, which explicitly display the trivial features with no edge modes.

Fig. S9. (a) Schematic of the orbital SSH model and the trivial numerical pressure eigen-fields. Noting that $\theta = 45^\circ$ is the smallest practical angle in this design. (b) The numerical energy spectra. The grey region denotes the unreachable bonding angles. (c) and (d) Alternative quasi-two-dimensional orbital SSH model and eigen-fields with arbitrary bonding angles.

Reviewer comment:

(6) *The dual symmetry looks interesting. In addition to its influence on energy dispersion, can it have other physical consequences?*

Response:

We agree that the hidden duality symmetry is very interesting. We have further discussed the influence of duality on the energy spectra and the topological edge states in hybridized orbital lattices composed of different bonding angles in the revised

manuscript. In Lines 198-200 (page 9), we have added “In fact, any orbital lattices with hybrid bonding angles θ and $2\theta_c - \theta$ possess the same energy spectra due to the hidden duality symmetry.”

In “**Supplementary Note 4**” of the revised Supplementary Information, we have added a paragraph and a new figure Fig. S7 to show the influence of duality symmetry on the topological properties of edge modes.

Related contents in **Supplementary Note 4** in the revised Supplementary Information: In addition, we have also explored other physical consequences endowed by the duality symmetry and performed the corresponding simulations. We have numerically calculated the eigenfrequency spectra and edge modes of the zigzag chains in two cases. One is a chain with the same bonding angle θ (Case1), and the other is a chain with θ on the left part and the corresponding dual one $2\theta_c - \theta$ on the right part (Case2). The results are shown in Fig. S7. As displayed in Figs. S7a-7c, all the eigenfrequencies for Case 1 and Case 2 are almost identical as θ are 70° , 75° and 80° , respectively. The duality symmetry protects the topological properties when the orbital system has the disorders obeying duality symmetry. The eigenfrequencies of edge modes clearly remain unchanged as we switch Case1 to Case2. In fact, any hybrid bonding angles obeying the duality symmetry do not change the eigenfrequency spectra. Furtherly, we have presented the simulated pressure fields of the orbital edge modes for the three hybrid lattices in Figs. S7d-7f, where the pressure fields are highly localized and polarized with p orbitals on the leftmost edge site with θ and the rightmost edge site with $2\theta_c - \theta$, showing a good agreement with the unitary lattice demonstrated in the main text.

Fig. S7. (a)-(c) The eigenfrequency spectra for the unitary and hybrid chains. (d)-(f) Corresponding topological edge modes, robust against the disorders obeying duality symmetry.

Reviewer #3

Reviewer comments:

In their manuscript "Orbital-Induced Acoustic Topological Order", Gao and collaborators investigate a one dimensional acoustic system of two coupled SSH chains. They realize this system with coupled acoustic resonators which support two modes, which are, in the absence of the coupling, degenerate. This allows to observe physics which is beyond the standard SSH model, i.e., results which are certainly worthy of publication. I have one main criticism on the presentation of the work which may influence the choice of the optimal journal for this manuscript. I find the physics observed interesting, albeit not exposed in enough detail, I comment on this below. What bothers me more is the claim to introduce a new paradigm for the design of interesting meta material band structures. The use of orbital degrees of freedom has always been a guiding principle to obtain the sought after features. Also in publications of some of the present authors. Having a mode with a node is a standard way to influence the (weak) coupling between coupled such modes. Moreover, using degenerate modes to engineer more complex band structures with non-trivial orbital

content is also not novel. For example in Nature Mat. 17, 323 (2018) a detailed recipe is given how two engineer tight-binding models exploiting orbital degrees of freedom, including the use degenerate pairs. In that sense, the current paper is an excellent illustration of these orbital based engineering, rather than a novel avenue. The fact that the above Nature Materials is formulated for waves in thin plates is irrelevant for the novelty. Both thin plates and acoustic cavities are described by classical wave equations. I would like to stress, however, that I consider the implemented model and observed phenomena of very high interest and novel. Whether this alone warrants a publication in Nature communications, I let the editors decide. My hunch would be that it is not quite ground-breaking enough for a journal like Nature Physics or Nature Materials, but may very well fit here. Before I can recommend the paper for publication I would invite the authors to comment on the following points:

Response:

We thank Reviewer #3 for these constructive comments and for their positive recommendation. In the following, we provide a point-to-point responses to their comments. We have also revised the manuscript and supplementary information following their suggestions.

Reviewer comments:

1.) Language: I find the term "topological order" unfortunate. In electron physics it is reserved for system with long-range entanglement like the fractional quantum Hall effect or certain spin liquids and not band effects. I would therefore avoid the use of "order" also in classical materials.

Response:

We thank Reviewer #3 for this insightful suggestion, and we agree that the use of the term “topological order” may not be fully appropriate in our context. To address this issue, we have altered the title “Orbital-Induced Acoustic Topological Order” to a more explicit one: “**Orbital topological edge states and phase transitions in one-dimensional acoustic resonator chains**” and replaced “topological order” in the main text by

“topological phase” in Lines 24, 83, 119 (pages 2, 4, 5), “topological effects” in Line 26 (page 2), “topological protection” in Line 37 (page 3), and “topological phase transitions” in Line 296 (page 14).

Reviewer comments:

2.) *Language: In line 39 (and elsewhere), the authors use the term "symmetry-breaking" DOF. I don't understand this term. Which symmetry is broken?*

Response:

We thank Reviewer #3 for pointing out this issue. We agree that the use of the term “symmetry-breaking” may not be precise and may cause confusion. We have revised the related sentence as “**Topological insulators are endowed by broken crystal symmetries and associated degrees of freedom (DoFs)**, such as spin or valley DoFs in a periodic lattice, playing an essential role in inducing various topological phases^{1,21}” in **Line 45-47 (page 3)** in the *revised manuscript*.

Reviewer comments:

3.) *Line 59 and 62-64 are not quite correct. Not only have higher-order resonances been considered before (e.g. mode 21 and 21 in the above mentioned Nature Mat.), but a much more detailed analysis of the nearby modes, their influence on the reduced order models an analysis of longer-range couplings, etc. I would simply remove these claims and devote more to the introduction of the actually studied model.*

Response:

We thank Reviewer #3 for this insightful suggestion, and we regret not noting this work before. Following the Reviewer’s suggestion, we have replaced the sentences in Lines 59-64 in the original manuscript with: “**Each resonator supports two degenerate orthogonal dipolar resonances in the frequency range of interest. A related concept has been used to design perturbative metamaterials in the elastic domain⁴², while the majority of topological acoustic research has been focusing on the use of single cavity resonances as building blocks⁴³⁻⁴⁹.**” in **Lines 72-76 (page 4)**.

Updated references:

42. Matlack, K. H., Serra-Garcia, M., Palermo, A., Huber, S. D. & Daraio, C. Designing perturbative metamaterials from discrete models. *Nat. Mater.* **17**, 323-328 (2018).

Reviewer comments:

4.) *In Fig. 1h one can see that the mean frequency of the sigma-coupled modes is a hunch lower than those of the pi-coupled bands, indicating that the coupling tubes between the cavities are not only acting as couplers but also influence them modes themselves. A careful analysis of this could be presented (not critical, however).*

Response:

We thank Reviewer #3 for this constructive comment. Yes, the parameters of the couplers not only influence the coupling efficiency, but also slightly shift the eigen-frequencies of the dimer units. To address this issue, we have performed simulations to analyze the impact on the σ -coupled modes and π -coupled modes as the coupling tube geometry is changed. The detailed results and analyses have been added in a new section “**Supplementary Note 2**” of the revised Supplementary Information:

Firstly, we explore the variation of eigenfrequencies and coupling strengths for dimer-units coupled with tubes of different diameters. We performed simulations to analyze the influence for σ -coupled modes and π -coupled modes when the diameter of the coupling tubes varies in the range [1 mm, 8 mm]. The obtained results are shown in Fig. S2. As shown in Fig. S2a, the eigen-frequencies of σ bonding (σ anti-bonding) and π bonding (π anti-bonding) both decrease (increase) as the diameter of the tube d varies from 1 mm to 8 mm. The eigenfrequencies of σ bonding (π bonding) are nearly symmetrical to the single-resonator eigenfrequency around 5028.5Hz.

Fig. S2. (a) Simulated eigen-frequencies for the dimer-units with the diameters of coupling tubes varying from 1 mm to 8 mm. The modes for σ bonding, π bonding, π anti-bonding, and σ anti-bonding are denoted by light blue, light green, green and blue spheres, respectively. (b) The mean frequencies of the σ -coupled modes and π -coupled modes. (c) The coupling coefficients for t_σ and t_π . (d) The contrast ratio.

The relations for the mean frequencies of the σ -coupled modes and π -coupled modes with the varying diameter d of coupling tubes are displayed in Fig. S2b. The mean frequency of the σ -coupled modes (black dotted line) and the π -coupled modes (orange dotted line) are almost identical as $d < 3$ mm. The mean frequency difference between the σ -coupled modes and π -coupled modes rapidly increases as $d > 3$ mm. In our simulations and experiments, we choose the diameter of the coupling tubes as $d = 6$ mm. The coupling strengths t_σ and t_π are proportional to half of the difference between the two related eigenfrequencies for the coupled dimer unit. The extracted effective coupling strengths have opposite signs: the values for t_σ and t_π are negative and positive, respectively³. The coupling coefficients t_σ and t_π vary with the diameter d , as exhibited in Fig. S2c. Clearly, the coupling strengths of t_σ and t_π increase as d varies from 1 mm to 8 mm, and t_σ increases faster than t_π . The absolute contrast ratio between t_π and t_σ is also plotted in Fig. S2d.

Reviewer comments:

5.) *Maybe a comment on the influence on the coupling strength of a shift of the two couples towards the node could be interesting.*

Response:

We thank Reviewer #3 for this insightful suggestion. Following this remark, we have explored and analyzed the influence on the coupling strengths of varied cavity intervals and coupler intervals. These results and analyses are summarized in “**Supplementary Note 2**” in the revised Supplementary Information.

Related content in **Supplementary Note 2** of the revised Supplementary Information: Next, we fix the diameter of coupling tubes to be $d = 6$ mm and investigate the influence of the interval between cavities or coupling tubes on the frequency spectra of the coupled resonators. In the first scenario, we vary the interval (w_2) between the dimer resonators (the length l of the paired coupling tubes varies accordingly), and perform simulations as l varied from 0 mm to 26 mm while fix the coupler interval as $w_1 = 16$ mm. Here we define $l = w_2 - D$ and $l = 0$ mm when the interval between two cavities is zero. The numerically calculated eigen-spectra for σ bonding, π bonding, π anti-bonding, and σ anti-bonding modes are shown in Fig. S3a. Clearly, the eigen-frequencies decrease as l varies from 0mm to 26mm. The coupling coefficients are furtherly depicted in Fig. S3b. t_σ gradually decreases as $l < 11$ mm, and then increases as $l > 11$ mm, implying a critical point at $l = 11$ mm. For t_π , the critical point is around $l = 8.5$ mm. The absolute contrast ratio of t_π / t_σ at each interval l is exhibited in Fig. S3c. For $l < 15$ mm, the contrast ratio varies very slowly but rapidly increases with l when $l > 15$ mm.

In the second scenario, we keep the interval between cavities $l = 13$ mm and vary the coupler interval parameter w_1 . For convenience, we set $w = w_1 / 2$ and investigate the variation of eigenfrequencies and coupling strengths for $w \in [3 \text{ mm}, 13 \text{ mm}]$. The numerically simulated eigen-spectra, coupling coefficients, and $|t_\pi / t_\sigma|$ are displayed in Figs. S3d-3f, respectively. The mode frequencies for σ bonding, π bonding, and σ anti-

bonding tend to decrease, while the mode frequency for π anti-bonding appears to increase, as shown in Fig. S3d. The absolute value of coupling coefficient t_σ decreases as w varies in the range [3 mm, 13 mm], while the value t_π keeps increasing, as displayed in Fig. S3e. Based on the above discussed, the coupling strengths for t_σ and t_π can be tailored by virtue of the variation of d , l and w , and the synergistic effect for the variation of l and w as well as d can be considered to obtain the coupling coefficient in demand.

Fig. S3. (a) Simulated eigen-frequencies for the dimer units with the interval (l) between two resonators varying in the range [0 mm, 26 mm]. The interval is 13 mm for the sample in the main text. The modes for σ bonding, π bonding, π anti-bonding, and σ anti-bonding are denoted by light blue, light green, green and blue spheres, respectively. (b) The coupling coefficients for t_σ and t_π . (c) The absolute contrast ratio between $|t_\pi/t_\sigma|$ with l . (d) Simulated eigen-spectra for the dimer units with varying w in the range [3 mm, 13 mm]. (e) The coupling coefficients of t_σ and t_π . (f) The absolute contrast ratio $|t_\pi/t_\sigma|$ with different w .

Reviewer comments:

6.) *Topology: The authors keep claiming these edge states to be topological. A careful*

analysis of this is certainly required to make such claims. Which is protecting symmetry? If SSH, it will be some chiral one. How is it represented in the space of the two modes in the two inequivalent cavities? How does this lead to a quantized Berry phase, or alike? Can I go to some block-off-diagonal representation as usual in this symmetry class? What is the influence of the duality on all of this?

Response:

We thank Reviewer #3 for pointing out these critical questions. Concerning the comments about *chiral symmetry and topological invariants* of our system, we propose a generalized chiral symmetry to describe its essential topological properties, of which the operator is defined as $\sigma_z \otimes \sigma_0$. Accordingly, the corresponding winding number characterizing the topological properties of the system can be extracted from the whole Hamiltonian (see Eq. (3) and Eq. (S16) below).

Concerning the comment about *block-off-diagonal representation*, we have explicitly shown the off-diagonal form in Eq. (2) of the main text, discussed the mapping of orbital-lattices model and conventional SSH models, and calculated their topological invariants in “**Supplementary Note 3**” of the *revised Supplementary Information*.

To address the above comments, we have implemented several changes in the revised manuscript: for example, we have added “The topologically protected edge modes are robust against defects and remain pinned to zero energy in a zigzag SSH chain with randomly selected bonding angles, **due to their topological protection associated with chiral symmetry³⁶**” in Line 94 (page 5), and “Each subspace corresponds to one copy of conventional SSH model and obeys the chiral symmetry.” In Lines 188, 189 (page 9), and “The winding number charactering the topological properties of the Hamiltonian can be defined as³⁶

$$\mathcal{W} = \frac{i}{2\pi} \int_{-\pi}^{\pi} dk \frac{d \ln \det D(k)}{dk} = -\frac{1}{2\pi} \int_C d \arg \det D(k) \quad (3)$$

where C represents a contour swept by $D(k)$ as k varies across the Brillouin zone.” in Lines 216-219 (page 10).

The contents relevant to the topological invariant and chiral symmetry in “**Supplementary Note 3**” of the revised Supplementary Information are pasted below: The topological properties of the orbital Hamiltonian are characterized by winding number defined by⁴

$$\mathcal{W} = \frac{i}{2\pi} \int_{-\pi}^{\pi} dk \frac{d \ln \det D(k)}{dk} = -\frac{1}{2\pi} \int_C d \arg \det D(k) \quad (\text{S16})$$

where C represents a contour swept by $D(k)$ as k varies across the Brillouin zone. For $|\theta - \pi/2| < \arcsin |(\gamma + 1)/(\gamma - 1)|$ where the system is gapped as shown in Fig. S5a, the contour of $\det D$ forms one turn around zero point, as shown in Fig. S5d, and thus we obtain the topological invariant $\mathcal{W} = 1$. For the critical condition at $|\theta - \pi/2| = \arcsin |(\gamma + 1)/(\gamma - 1)|$, the contour for $\det D$ is displayed in Fig. S5e, and the winding number is ill-defined. When the band is gapless as shown in Fig. S5c, $\det D$ actually passes twice through zero point defined by $\det D = 0$, which is shown in Fig. S5f. To formally define the winding number, the contour in Fig. S5f can be shifted by an infinitesimal amount to the left or to the right, for which we obtain the topological invariant $\mathcal{W} = 0$.

Fig. S5. Calculated and simulated energy spectra marked by black lines and red spheres for different bonding angles at (a) 100° , (b) 132.9° , and (c) 165° . Schematic sketches for a contour C in the complex plane that $\det D$ follows as k changes from $-\pi$ to π .

(d) Scenario for $|\theta - \pi/2| < \arcsin |(\gamma+1)/(\gamma-1)|$. (e) Scenario for $|\theta - \pi/2| = \arcsin |(\gamma+1)/(\gamma-1)|$. (f) Scenario for $|\theta - \pi/2| > \arcsin |(\gamma+1)/(\gamma-1)|$.

For the topological invariants of the orbital SSH model, we can also illustrate the winding numbers from the conventional SSH models⁵. A standard SSH Hamiltonian reads

$$H_{SSH} = \sum_n (t a_n b_n^\dagger + t' a_n^\dagger b_n) + h.c. \quad (\text{S17})$$

where $t(t')$ represents the intracell (intercell) hopping amplitude, and $a_n^\dagger(b_n^\dagger)$ is the creation operator on the site $a_n(b_n)$ in the n th unit cell. H_{SSH} displays two topologically distinct phases as two different dimerizations $t > t'$ and $t < t'$ are considered. The different topology of the two phases is unveiled by considering the winding \mathcal{W} of phase $\phi(k)$ across the Brillouin zone:

$$\mathcal{W} = \frac{1}{2\pi} \int_{BZ} \frac{\partial \phi(k)}{\partial k} dk \quad (\text{S18})$$

which corresponds to the Zak phase divided by π . For $t > t'$, the intracell coupling is stronger than intercell coupling, as shown in Fig. S6a, corresponding to the trivial case with $\mathcal{W} = 0$. When we cut the lattice into a finite array and keep the stronger couplings on the end, no edge modes will appear. In contrast, for $t < t'$ depicted in Fig. S6b, we anticipate nontrivial edge modes on the ends with weaker couplings and the winding number $\mathcal{W} = 1$.

For the zigzag chain in Fig 2a of the main text, the couplings are subjected to alternating strengths along the chain for p_x or p_y orbital, each of which corresponds to one copy of conventional SSH model. In the zigzag arrays with bonding angle $\theta = 90^\circ$ shown in Figs. S6c and 6d, p_x and p_y orbitals are separately oriented along the diagonal and antidiagonal axes, and the hopping amplitude between consecutive resonators strongly relies on the orientation of the axis connecting the resonators. The hopping strengths longitudinal and transverse to the bond between the cavities are denoted as

t_σ and t_π , respectively, and the magnitude of t_σ is larger than that of t_π in our design. Then, as the subspace of p_x mode is excited (Fig. S6c), effective dimerized coupling pattern emerges along the chain. In this subspace, effective stronger coupling t_σ ($t_\sigma > t_\pi$) appears on the end, and no edge states are expected to emerge (Fig. S6c), which corresponds to the trivial topological phase ($\mathcal{W}=0$). For the p_y subspace depicted in Fig. S6d, it terminates with weaker coupling t_π , and topological edge states are anticipated on the boundaries, corresponding to the nontrivial topological phase ($\mathcal{W}=1$).

Fig. S6. (a) and (b) Schematic sketches for the two different dimerizations in the conventional SSH models. (c) and (d) Schematic sketches corresponding to the p_x and the p_y subspaces of the orbital SSH models. Each subspace displays similar coupling arrangements as the conventional SSH models.

S3.3 Chiral symmetry in the orbital SSH models

For the orbital SSH models with bonding angle $\theta = 90^\circ$ shown in Figs. S6c and S6d, they also share chiral symmetry as the conventional ones^{5,6}. There exists a unitary transformation U_c that anticommutes with the Hamiltonian: $\{H, U_c\} = 0$. For the p_x/p_y subspace SSH lattice with bonding angle $\theta = 90^\circ$ as discussed above, the Hamiltonian of the system (Eq. (S17)) can be expressed in the momentum space

$$H(k) = \vec{d}(k) \cdot \vec{\sigma}, \quad (\text{S19})$$

where $\sigma_{x,y,z}$ are the Pauli matrices, and

$$d_x(k) = t + t' \cos(ka), d_y(k) = t' \sin(ka), d_z(k) = 0 \quad (\text{S20})$$

with a being the unit-cell constant⁷. The chiral operator is defined by the Pauli matrix σ_z , namely $\{H, \sigma_z\} = 0$. Thus, the chiral symmetry of orbital SSH Hamiltonian is preserved. On the other hand, for the general orbital Hamiltonians with arbitrary bonding angles as described by Eq. (2) in the main text, it also obeys the generalized chiral symmetry as $\{H, \sigma_z \otimes \sigma_0\} = 0$.

Concerning the comment about *the influence of the duality*, we have added several sentences: “In fact, any orbital lattices with hybrid bonding angles θ and $2\theta_c - \theta$ possess the same energy spectra due to the hidden duality symmetry.” in Lines 198-200 (page 9) and “In our sample, the bonding angle couldn’t be smaller than 45° and the related physics can be explored in alternative samples with bonding angles between 135° and 180° due to the hidden duality symmetry.” in Lines 209-211 (page 9) in the revised manuscript.

Moreover, we have added a long paragraph and a new figure Fig. S7 to show the influence of duality symmetry on edge modes in “**Supplementary Note 4**” in the revised Supplementary Information:

In addition, we have also explored other physical consequences endowed by the duality symmetry and performed the corresponding simulations. We have numerically calculated the eigenfrequency spectra and edge modes of the zigzag chains in two cases. One is a chain with the same bonding angle θ (Case1), and the other is a chain with θ

on the left part and the corresponding dual one $2\theta_c - \theta$ on the right part (Case2). The results are shown in Fig. S7. As displayed in Figs. S7a-7c, all the eigenfrequencies for Case 1 and Case 2 are almost identical as θ are 70° , 75° and 80° , respectively. The duality symmetry protects the topological properties when the orbital system has the disorders obeying duality symmetry. The eigenfrequencies of edge modes clearly remain unchanged as we switch Case1 to Case2. In fact, any hybrid bonding angles obeying the duality symmetry do not change the eigenfrequency spectra. Furtherly, we have presented the simulated pressure fields of the orbital edge modes for the three hybrid lattices in Figs. S7d-7f, where the pressure fields are highly localized and polarized with p orbitals on the leftmost edge site with θ and the rightmost edge site with $2\theta_c - \theta$, showing a good agreement with the unitary lattice demonstrated in the main text.

Fig. S7. (a)-(c) The eigenfrequency spectra for the unitary and hybrid chains. (d)-(f) Corresponding topological edge modes, robust against the disorders obeying duality symmetry.

Additionally, to furtherly illustrate topological robustness, we considered on-site resonant frequency perturbations. We have added the numerical results and analyses in “**Supplementary Note 7**” in the revised Supplementary Information:

To further illustrate the robustness, the on-site resonant frequency perturbations are

taken into consideration, which could be the major origins of disorders in experiments because of the fabrication errors. Here, two scenarios are considered: one is that disorders are on all the sites except the left and right edge site (Scenario 1), and the other one is that disorders are only introduced on the two edge sites (Scenario 2). We perform simulations for these two scenarios. Small hard cylinders whose heights are $h_c = 2$ mm and radii r_c are randomly distributed in the range $[0.5\text{mm}, 2.5\text{mm}]$, are placed at the bottom of the resonators⁸ (shown in Fig. S12a). We separately simulate energy spectra of the zigzag chain by 50 random cases for both Scenario 1 and Scenario 2, which are shown in Figs. S12b and S12c. For Scenario 1, the eigenfrequencies of edge states clearly keep invariant in the bandgap (Fig. S12b), while for scenario 2, the edge states' eigenfrequencies are fluctuated (Figs. S12c). We furtherly display the spectra of specific lattice configurations as marked by the blue arrows in Figs. S12b and S12c, as shown in Figs. S12d and S12e. The eigenfrequencies of edge states obviously remain stable and degenerate in Scenario 1. However, in Scenario 2, the edge states still exist but the eigenfrequencies are no longer degenerate. Figure. S12f shows the simulated response spectra at one edge resonator in different scenarios. For Scenario 1 (bulk disorder), the resonance peak remains unchanged, while for Scenario 2 (edge disorder), the peak still exists but shifts to a lower frequency.

Fig. S12. (a) Schematic of the disorder introduced in the resonator. A small rigid cylinder (blue) is put at the bottom of the resonator. (b), (c) Energy spectra for 50

different lattices with rigid cylinders of random radii in the resonators, introducing the bulk disorder and edge disorder, respectively. (d), (e) The eigenfrequency spectra for specific lattice configurations marked by blue arrows in (b), (c) respectively. (f) Simulated response spectra at one edge for the chain with no disorder (dark blue line), bulk disorder (red line) and edge disorder (black line).

REVIEWERS' COMMENTS

Reviewer #2 (Remarks to the Author):

The manuscript has been thoroughly revised. I can now recommend publication.

Response Letter to Reviewers

We sincerely thank the reviewers for their careful review of the manuscript and their constructive guidance, which were crucial to improving our manuscript. We thank the reviewers for their patience and careful review.

Reply to the Reviewer #2

Main comments:

The manuscript has been thoroughly revised. I can now recommend publication.

Response:

We thank the reviewers for their careful review and recommendation.